# Effects of Loading Forces, Loading Positions, and Splinting of Two, Three, or Four Ti-Zr (Roxolid^®^) Mini-Implants Supporting the Mandibular Overdentures on Peri-Implant and Posterior Edentulous Area Strains

**DOI:** 10.3390/jfb15090260

**Published:** 2024-09-09

**Authors:** Nikola Petricevic, Asja Celebic, Dario Puljic, Ognjen Milat, Alan Divjak, Ines Kovacic

**Affiliations:** 1Department of Removable Prosthodontics, School of Dental Medicine, University of Zagreb, 10000 Zagreb, Croatia; dpuljic@sfzg.hr (D.P.); kovacic@sfzg.hr (I.K.); 2Institute of Physics, 10000 Zagreb, Croatia; milat@ifs.hr; 3Department of Digital Arts, University Algebra, 10000 Zagreb, Croatia; alan.divjak@algebra.hr

**Keywords:** Ti-Zr mini-implants, strain gauges, mandibular overdenture, loading forces, number of implants, splinting, peri-implant strains, edentulous area strains

## Abstract

Clinical indications for the Ti-Zr alloy (Roxolid^®^) mini-implants (MDIs) in subjects with narrow ridges are still under review. The aim was to analyze peri-implant and posterior edentulous area strains dependent on the MDI number, splinting status, loading force, and loading position. Six models were digitally designed and printed. Two, three, or four Ti-Zr MDIs, splinted with a bar or unsplinted (single units), supported mandibular overdentures (ODs), loaded with 50–300 N forces unilaterally, bilaterally, and anteriorly. The artificial mucosa thickness was 2 mm. Strain gauges were bonded on the vestibular and oral peri-implant sides of each MDI, and on the posterior edentulous area under the ODs. Loadings were performed through the metal plate placed on ODs’ artificial teeth (15 times repeated). Arithmetic means with standard deviations and the significance of the differences (MANOVA, Sheffe *post hoc*) were calculated. Different MDI numbers, loading positions, forces, and splinting elicited different peri-implant microstrains. In the two-MDI models, 300 N force during unilateral loading elicited the highest microstrains (almost 3000 εμ on the loaded side), which can jeopardize bone reparation. On the opposite side, >2500 εμ was registered, which represents high strains. During bilateral loadings, microstrains hardly exceeded 2000 εμ, indicating that bilateral chewers or subjects having lower forces can benefit from the two Ti-Zr MDIs, irrespective of splinting. However, in subjects chewing unilaterally, and inducing higher forces (natural teeth antagonists), or bruxers, only two MDIs may not be sufficient to support the OD. By increasing implant numbers, peri-implant strains decrease in both splinted and single-unit MDI models, far beyond values that can interfere with bone reparation, indicating that splinting is not necessary. When the positions of the loading forces are closer to the implant, higher peri-implant strains are induced. Regarding the distal edentulous area, microstrains reached 2000 εμ only during unilateral loadings in the two-MDI models, and all other strains were lower, below 1500 εμ, confirming that implant-supported overdentures do not lead to edentulous ridge atrophy.

## 1. Introduction

A lack of mandibular complete denture retention in subjects with atrophied residual ridges of limited width can often be successfully solved in a simple way by the insertion of four Ti90Al6V4 alloy one-piece mini-dental implants (MDIs) in the interforaminal region, either by an open-flap or a flapless surgical approach [1,2,3,4,5,6,7,8,9,10,11,12,13]. The MDIs are listed in category 1 of narrow implants with a diameter equal to or less than 2.5 mm [14]. Many longitudinal clinical studies proved that such simplified protocols without augmentation procedures, together with the possibility of immediate loading, have positive effects, manifested by increased patient satisfaction, improved oral function, good peri-implant health, and satisfactory success and survival rates of MDIs [3,4,7,9,11,15,16,17,18,19]. However, some studies reported lower survival rates of MDIs when compared to standard-diameter implants [20], while some studies reported the 100% survival rates of MDIs [3,4,11].

Advances in the materials for dental implant construction have led to the development of a high-strength (grade 5) titanium and zirconium alpha alloy with excellent osseointegration, which the Straumann group patented as the Roxilid^®^ alloy (Ti85Zr15) almost ten years ago [21,22,23,24,25]. Clinical longitudinal studies reported that the Roxilid^®^ alloy two-piece narrow (category 2 or 3) dental implants have high success rates [26,27].

A few years ago, the Roxilid^®^ alloy as well as some other innovations were implemented in the new Straumann^®^ Mini-Implant System^®^, with the aim to improve clinical predictability of MDIs when stabilizing overdentures [28]. The Straumann^®^ Mini-Implant System^®^ represents the 2.4 mm wide (category 1 of narrow implants) one-piece implant, made of the Roxolid^®^ alloy with a sandblasted, large-grit, acid-etched implant surface (SLA^®^). The Straumann^®^ mini-implant tapers apically, allowing surgical under-preparation, similar to other mini-implant systems. The new Ti-Zr mini-implant system has a male prosthetic connection in a form of a rounded head emerging on the small platform from keratinized oral mucosa of a denture bearing area. It is coated with an amorphous diamond-like carbon (ADLC). The ADCL has a task to enhance durability of the male part of the prosthetic connection. The female part consists of a high-performance PEEK (Polyether ether ketone) matrix, which is placed in the titanium metal housing (Straumann^®^ Optiloc Retentive System^®^). Studies that were focused on mandibular overdentures supported by four Straumann^®^ mini-implants covering one year of clinical follow-up reported high implant survival and success rates, improvements in overdenture retention and oral function, low surgical burden, good peri-implant health, and satisfactory prosthetic outcomes [29,30,31].

Due to good performance of the Straumann^®^ Mini-Implant System^®^, but also due to high costs of the recommended four mini-implants (preventing their mass utilization), “in vitro” studies were conducted to determine whether fewer than four Ti-Zr Straumann^®^ mini-implants can be used for mandibular overdenture support [32,33]. The mandibular overdentures were loaded with forces ranging from 50 to 150 N. Peri-implant strains increased while increasing the extent of loading forces and decreasing the number of implants. Even when two MDIs were splinted, relatively high peri-implant strains were registered during unilateral loadings. It was concluded that precaution and additional study should be addressed when only two Ti-Zr MDIs support mandibular ODs.

The aims of the present study were to measure peri-implant and edentulous area strains under higher loading forces up to 300 N, as well as to analyze effects of splinting of two, three, or four Ti-Zr mini-implants.

## 2. Materials and Methods

### 2.1. Preparation of Models and Overdentures

A detailed description of all materials and the experimental schedule has already been reported [32,33]. The models of the mandible were based on the CBCT scans of one convenient edentulous patient with an atrophied alveolar ridge of reduced width. We designed the virtual mandibular model (CAD, Amira software, v4.1, Zuse Institute Berlin; Visage Imaging GmbH, Berlin, Germany). In the virtual model, the positions for the MDIs were planned and designed (the Blender^®^ software, v2.79b, Amsterdam, The Netherlands). Each virtually designed mandibular model was created with a different number of holes (two, three, or four holes). Each hole was 1 mm narrower than the Ti-Zr mini-implant width. In the two models, four holes were created for the insertion of four MDIs (in positions of previous first premolars and second incisors on the right and left sides of the mandible). In another two models, the holes for the insertion of three MDIs were created (two posterior MDIs in the positions of previous distoproximal surfaces of the right and left canines, and one anterior MDI in the midline of the mandible). The last two models were designed with the two holes (positions of the previous left and right mandibular canines) for the insertion of two Ti-Zr mini-implants. The stereolithographic 3D printing technology (Form 2, Formlabs, Somerville, MA, USA) and the Gray photopolymer resin (GRAY FLGPGR04; Formlabs, Somerville, MA, USA) were used to print all the models, which were further processed as follows: The models were immersed in the 95% isopropyl alcohol (IPA) (Izopropil alcohol, Medimon d.o.o., Split, Croatia) throughout one minute and after that, the models were additionally immersed during 15 min in a new container of IPA in order to rinse the residual resin. After cleaning, the polymerization with the 36 W UV-A halogen lights was performed during a period of 30 min (Dentsply Sirona Heliodent Plus, Display Sirona, York, PA, USA) and after that, the 30 min heating in a chamber at 60 °C was performed for each model. The material of the models mimicked the D2 bone density. Each model was covered with an artificial mucosa, which was made from vinyl-polysiloxane impression material (3M™ Express™ XT Light Body Quick, Seefeld, Germany). The artificial mucosa had a uniform thickness of 2 mm. In each hole, which was 2.3 mm wide and 10 mm long, the mini-implant, which was 2.4 mm wide and 10 mm long (Straumann^®^ Mini-Implant, Institute Straumann AG, Basel, Switzerland) with a neck height of 2.8 mm, was inserted. The diameter of the hole was 0.1 mm narrower than the implant width, while the length was the same as the implant length. The original surgical kit was used for mini-implant insertion. The insertion torque varied only for a small amount (5 Ncm) between the sites of insertion with values >35 Ncm in all sites. In the three models of the mandible (with two, three, and four implants, respectively), the mini-implants were splinted with a bar designed in the 3Shape software (3Shape, Copenhagen, Denmark). After the bar was designed, it was milled from BEGO Mediloy^®^ M-Co (BEGO, Bremen, Germany) using the Imes-icore CORiTEC 350i machine (GmbH, Germany). After milling and polishing, the bars were cemented on the implant necks. The adhesive cement was used for the bar cementation (Maxcem Elite^TM^ Self-Etch/Self Adhesive Resin Cement, KaVo Kerr, Brea, CA, USA). The other three models remained with MDIs as single units, without splinting. Being briefly summarized, in the three models (two, three, or four implants each), MDIs were splinted, while in another three models, mini-implants supported the overdentures as single units (Figure 1a–c). The models were scanned with the laboratory scanner (3Shape 3E, 3Shape, Copenhagen, Denmark) to proceed with the overdenture manufacturing. Metal frameworks for the overdenture were designed using computer-aided design (CAD) technology in the 3Shape software (3Shape, Copenhagen, Denmark), and printed from Wironium^®^ RP metal powder (BEGO, Bremen, Germany) with a Sisma Mysint100 Dual laser (Sisma, Piovene Rocchette, Italy). Frameworks were incorporated into the overdentures made in the laboratory from the modeling wax. The artificial teeth (Ivostar, Ivoclar Vivadent, Schaan, Liechtenstein) were used and set into the wax. The overdentures were processed according to the manufacturer’s recommendation (Ivoclar ProBase Hot Denture Resin, Ivoclar Vivadent, Schaan, Liechtenstein) and the wax was replaced with a resin. After processing, the overdentures were polished. The medium (yellow) PEEK retention inserts (1200 g retention force) were placed into the titanium housings, which were built into the overdenture in the single-unit MDI models (two, three, or four MDIs each). In the MDI models, which were splinted with the bar (two-, three-, or four-MDI models), the ready-made yellow plastic clips were used (CEKA, PRECI-HORIX COMBI, Waregem, Belgium). The plastic clips attached the overdentures to the bars.

### 2.2. Strain Gauge Mounting and Measurements

For peri-implant and posterior edentulous area strain measurement, the strain gauges (SGs) (KFGS-1N-120-C1-11N30C2, Kyowa Electronic Instruments Co., Ltd., Tokyo, Japan) were used. They were adhesively cemented on the models. Before the adhesive cementation, the surface of each model was cleaned with acetone before the application of the cyanoacrylate glue (Super Glue, NU Co. Ltd., Ningbo, China). The acetate foil (Grafix Clear Acetate, Grafix^®^, Plastics, Maple Heights, OH, USA) was used for pressing the strain gauges on the vestibular and oral peri-implant surfaces as close as possible to each mini-implant during the cementations (Figure 1a). To obtain strains elicited by different loadings in the posterior edentulous area under the overdenture, the SGs were glued near the posterior end of the free-end overdenture saddles in the same way using the acetate foil (Figure 1a).

To measure strains elicited by the ODs’ loadings, the recording system EDX-10A, Kyowa Electronic Instruments Co., Ltd., Tokyo, Japan, was used. During the recordings, all strain gauges were connected to the software (DCS-100A, Kyowa Electronic Instruments Co., Ltd., Tokyo, Japan), which allowed for simultaneous monitoring and recording of the deformations, which were elicited by the respective overdenture loadings (Figure 2a). During the denture loadings, each model was fixed in a special stand with two round bars placed horizontally to support the model. The metal screw touching the metal plate with the rounded end was twisted to apply a pressure on the metal plate, which was placed over the overdenture’s artificial teeth, transferring the applied loads to the respective overdenture (Figure 2a). The overdentures were loaded in the three different positions: bilaterally, unilaterally, and anteriorly (Figure 2a–d). During bilateral loadings, the metal plate was positioned over the second premolars and the first molar teeth, while the metal screw with the balled end was also connected at the same time to a force measuring cell.

During the unilateral overdenture loadings (only on the right side), the metal plate was positioned unilaterally on the artificial premolars and the first molar of the respective overdenture (Figure 2c), and during anterior loadings on the denture’s artificial incisors (representing anterior loadings) (Figure 2d). Loading forces were 50 N, 100 N, 150 N, 200 N, 250 N, and 300 N, respectively, during bilateral and unilateral loadings, and 50–250 N during anterior loadings. Microstrains were registered from the vestibular and oral SGs in peri-implant sites, and from posterior edentulous areas under the OD saddles. For the registration of strains from the posterior edentulous areas, only bilateral and unilateral loadings were performed. Loadings were performed during the interval of a few seconds until the desired loading force was achieved, which was then maintained for 2 s. The highest value during this 2 s interval was chosen for the statistical analysis. Briefly, all measurements were repeated 15 times, and the maximum values of each of the 15 measurements for each loading position and each force were entered for the statistical analysis.

### 2.3. Statistical Analysis

To obtain the statistical analysis, the SPSS 22 software was used. The normality of the distribution was confirmed by the one-sample Kolmogorov–Smirnov test. Mean values and standard deviations for each set of fifteen measurements were calculated. The microstrain values are presented in graphs as estimated marginal means. The three-factor MANOVA and the Sheffe post hoc tests (for a comparison of more than two groups) were used to obtain the significance of the differences of the peri-implant microstrains depending on the extent of applied forces (50–300 N), loading position (bilateral, unilateral, anterior), and splinting status (single-unit unsplinted MDI models or splinted MDI models with a bar) in the MDI models (with two, three, or four MDIs inserted, respectively). The four-factor MANOVA and the Scheffe post hoc tests were used to obtain the significance of the differences of the microstrains registered from the posterior edentulous areas depending on the extent of applied forces (50–300 N), loading position (unilateral or bilateral), splinting status (splinted vs. single-unit MDIs), and number of mini-implants inserted (two, three, and four MDIs). The effect sizes were assessed by the partial eta-squared and interpreted as follows: η^2^ = 0.01 was a small effect, η^2^ from 0.06 to <0.14 indicated a medium effect, and η^2^ = 0.14 indicated a large effect.

## 3. Results

### 3.1. Peri-Implant Microstrains

#### 3.1.1. Two Mini-Implants Supporting Mandibular Overdentures

Microstrains registered from the vestibular and oral peri-implant strain gauges in the two-mini-implant models, both when single-unit or splinted mini-implants supported the mandibular overdenture, which was loaded bilaterally, anteriorly, and unilaterally (right side) with loading forces of 50, 100, 150, 200, 250, and 300 N, are presented graphically in Figure 3 as estimated marginal means. Descriptive statistics (mean values and standard deviations) is presented in Appendix A.

The highest peri-implant microstrains were obtained from the right-side vestibular and oral SGs when the OD was loaded unilaterally (on the right side) with the 300 N force, followed by the 250 N force. Peri-implant microstrains almost reached 3000 εμ, which could jeopardize bone reparatory mechanisms in a real patient situation [34,35]. Peri-implant strains recorded from the left SGs during unilateral (right-side) loadings were a bit lower than on the right side and reached 2500 εμ. Lower peri-implant microstrains were registered under lower forces, bilateral and anterior loadings, while the lowest microstrains were recorded under the lowest 50 N loading forces. During the OD loadings, by increasing the loading forces, peri-implant strains increased almost linearly.

The multivariate analysis with peri-implant microstrains as dependent variables and loading forces, loading positions, and splinting status as factors is presented in Appendix A. The extent of applied force, loading position, and splinting status showed significant effects (*p* < 0.001), but Force* Splinting status and Loading position*Force did not show significant effects (*p* > 0.05). Post hoc Scheffe tests for the independent variable force in the two-MDI models are presented in Appendix A. By increasing the extent of force applied to the mandibular OD, peri-implant microstrains also increased (*p* < 0.01), almost in a linear manner. The post hoc tests (Sheffe) for the independent variable loading position in the two-MDI models are presented in Appendix A. Peri-implant microstrains during loading of the mandibular OD in the three loading positions (bilaterally, anteriorly, and unilaterally on the right side) were significantly different from each other, except for the peri-implant microstrains from the left oral SG when strains during bilateral loadings did not differ significantly from anterior loadings. On the left side, under the 150 N force, there was a significant difference between the “Not Splinted (Single Units)” and “Splinted” models considering the registered microstrain values during anterior and unilateral loadings.

#### 3.1.2. Three Mini-Implants Supporting the Mandibular Overdentures

The peri-implant microstrains registered from the vestibular and oral strain gauges in the three-mini-implant models (the right, left, and midline MDIs), when mini-implants (single-unit or splinted) supported the mandibular ODs loaded bilaterally, anteriorly, and unilaterally (on the right side) with forces of 50 N, 100 N, 150 N, 200 N, 250, and 300 N, respectively, are presented in Figure 4. Descriptive statistics, i.e., mean values and standard deviations, is presented in Appendix A. Peri-implant microstrains in the three-MDI models were lower than in the two-MDI models under the same conditions.

The highest peri-implant microstrains were registered from the right-side MDI SGs when the OD was loaded unilaterally (on the right side) with the 300 N force. It did not exceed 2400 εμ. A bit smaller peri-implant microstrains were registered under 250 and 200 N forces. Lower microstrains were recorded under lower forces, bilateral and anterior loads, while the lowest peri-implant microstrains were recorded under 50 N loading force and anterior loadings in the right and left MDIs. The midline MDI also showed relatively high peri-implant microstrains under anterior loads, but microstrains did not exceed 2000 εμ (under the highest 250 N anterior loading force).

The multivariate analysis with peri-implant microstrains as dependent variables and loading forces, loading positions, and splinting status as factors in the three-MDI models is presented in Appendix A. Like in the two-MDI models, the extent of applied force, loading position, and splinting status showed significant effects (*p* < 0.001). However, Force* Splinting status and Loading position*Splinting status did not show significant effects (*p* > 0.05). Force*Splinting Status*Loading Position also did not show significant effects.

The post hoc tests for the independent variable force in the three-MDI models are presented in Appendix A. By increasing the extent of forces applied to the mandibular OD, the peri-implant microstrains also increased (*p* < 0.01). Each force (50, 100, 150, 200, 250, and 300 N, respectively) elicited peri-implant microstrains, which were significantly different from each other. The post hoc tests for the independent variable loading position in the three-MDI models are presented in Appendix A. Peri-implant strains were significantly different dependent on the loading positions (bilateral, anterior, unilateral–right side) with unilateral–right side loadings eliciting the highest peri-implant microstrains.

#### 3.1.3. Four Mini-Implants Supporting the Mandibular Overdentures

Microstrains registered from the vestibular and oral peri-implant strain gauges in the four-mini-implant models (either single-unit or splinted) (right and left posterior and right and left anterior MDIs), when the mandibular OD was loaded bilaterally, anteriorly, and unilaterally (right side) with loading forces of 50 N, 100 N, 150 N, 200 N, 250, and 300 N, respectively, are presented in Figure 5. Descriptive statistics (mean values and standard deviations) is presented in Appendix A. Generally, peri-implant microstrains were lower than in the three-MDI models under the same conditions.

The multivariate analysis of peri-implant microstrains (dependent variable) in the four-MDI model dependent on the loading force, splinting status, and loading position is presented in Appendix A. All factors elicited significant effects (*p* < 0.01). The post hoc tests for the independent variable force in the four-MDI models are presented in Appendix A. By increasing the extent of forces applied to mandibular ODs, peri-implant microstrains also increased. Peri-implant microstrains elicited by each of the applied forces (50, 100, 150, 200, 250, and 300 N, respectively) were significantly different from each other (*p* < 0.01). The post hoc tests for the independent variable loading position in the four-MDI models are presented in Appendix A. The peri-implant microstrains were significantly different in each loading position (bilateral, anterior, unilateral–right side) in the four-MDI model (*p* < 0.01), again with the highest values recorded from the peri-implant SGs of the posterior MDIs during unilateral loadings, which were about 2000 εμ, while all other peri-implant strains were lower.

### 3.2. Microstrains Registered from the Posterior Edentulous Areas

Microstrains registered from the right and left posterior edentulous areas under the mandibular overdenture supported by single-unit or splinted MDIs in the two-, three-, and four-MDI models when the overdentures were loaded in different loading positions (unilaterally, bilaterally, and anteriorly) with different loading forces (50, 100, 150, 200, 250, and 300 N, respectively) are presented in Figure 6. Descriptive statistics of the microstrains in the two-MDI models (single-unit or splinted) registered from the right-side and the left-side posterior edentulous areas (under the mandibular overdenture) is presented in Appendix A. Means and standard deviations of the microstrains registered from the posterior edentulous areas in the three-MDI models (single-unit or splinted) are shown in Appendix A, while means and standard deviations of microstrains registered from the four-MDI models (single-unit or splinted) are shown in Appendix A.

Microstrains recorded in the posterior edentulous area generally decreased when increasing the number of mini-implants or decreasing the loading forces. The highest microstrain values (a bit lower than 2000 εμ) were recorded from the right-side edentulous area in the single-unit two-MDI model during unilateral loading (on the right side) with the highest 300 N forces. The splinting of mini-implants elicited a bit lower strains in the edentulous areas, especially under higher forces. The multivariate analysis with the posterior edentulous area microstrains (on the right side and on the left side) as dependent variables and number of MDIs, splinting status, loading position, and loading force as factors elicited significant effects (*p* < 0.01), which are presented in Appendix A. The posterior edentulous area microstrains were significantly higher under unilateral (right side) than under bilateral loadings (*p* < 0.01). Splinted status elicited significant effects. The microstrains obtained from the posterior edentulous areas under the OD saddles were smaller in the splinted models, especially under higher forces. The post hoc Sheffe tests depending on the variable force for the posterior edentulous area microstrains are presented in Appendix A. Microstrains were significantly different under different forces and were increasing as the loading force increased (*p* < 0.01). The post hoc Sheffe tests dependent on the number of mini-implants, presented in Appendix A, showed that microstrains in the edentulous areas decreased significantly by increasing the number of implants (*p* < 0.01); however, no significant difference was registered between the three- and four-mini-implant models on the left side.

## 4. Discussion

In vitro studies are very important before the implementation of new methods and materials in clinical work, especially in the research of oral implants, as they provide important information about implant behavior under masticatory forces. By measuring peri-implant stress and strains, as well as stress and strains from the posterior edentulous areas under the overdenture, it is possible to predict a possibility of peri-implant marginal bone loss, or bone atrophy under overdenture saddles, or even an implant fracture [34,35,36]. Approximately 1000 microstrains correspond to 0.1% deformation in bone, which vary for a small amount among different bone densities [37]. By increasing stress and strains, the bone first compensates by the formation of a new bone and can function without damage within the range of 50–2500 microstrains. However, repeated stresses (>2000 µε) and/or microstrains equal to or above 3000 µε increase the micro-damage in bone and interfere with bone reparatory mechanisms [34,35,37]. Our previous studies [32,33], when overdentures were loaded with forces varying from 50 to 150 N, which represent average chewing forces in subjects wearing implant-supported overdentures [38,39,40], revealed that under unilateral loading conditions and 150 N forces, peri-implant strains reached 2000 µε in the models with two single-unit or splinted MDIs, as well as in the one-MDI model, representing mild to average overloads. If stresses are repeated frequently, especially in older subjects with co-morbidities, taking multiple medications, or having lower bone densities; in heavy smokers; or in subjects with any medical conditions that slow bone turnover [41,42,43,44,45], peri-implant marginal bone loss could perhaps begin, or the resorption of bone in edentulous areas under the denture saddles. In bruxers and in individuals having natural teeth or fixed partial dentures in the antagonistic jaw, chewing forces may be higher than the average forces in overdenture patients (150 N) [46,47,48,49] and may elicit high peri-implant strains at the implant–bone interface and marginal bone loss. In the previous studies [32,33], it was postulated that additional investigations should be conducted by loading the overdentures with higher forces, which might elicit higher peri-implant strains. Therefore, it was one of the aims of this study. The previous study also did not show significant effects of splinting on peri-implant strains in the two-MDI model under loading forces up to 150 N [33]; however, behavior of both splinted and unsplinted MDIs under higher forces is not known, and therefore it is an aim of the present study. The aim is also to observe the effect of splinting in the two-, three-, and four-MDI models. Measuring strains in peri-implant bone in narrow implants is very important as a decrease in implant diameter increases stresses in the implant bone interface [50].

All models in this study represented the same mandible (except for the different number of holes for MDI insertion), while the material of the models mimicked the D2 bone density, which is most frequently found in the interforaminal region of the mandible [51]. To approximate flat surfaces, the narrowest strain gauges of 1 mm were used because the inaccuracies caused by anatomical factors were minimized. Moreover, a firm pressure was applied on the acetate foil placed over strain gauges during cementation, so that only a thin layer of adhesive remained to minimize possible errors due to variations in the thicknesses of the bonding agent [52]. Bilateral and unilateral loadings were performed with forces up to 300 N, while anterior loadings were performed with forces up to 250 N. Lower anterior loadings up to 250 N were chosen because occlusal forces decrease in the incisor region and increase in the premolar and molar regions [53].

Generally, peri-implant microstrains increased almost in a linear manner when loading forces were increased in all models (two, three, or four MDIs). The peri-implant strains were the highest during unilateral overdenture loadings with the highest forces in the two-MDI models. Peri-implant microstrain values almost reached 3000 εμ on the loaded side during unilateral loadings in the two-MDI models (both single-unit and splinted) and were higher than on the opposite side, and in the splinted model compared to the single-unit model. Peri-implant strains recorded from the left side (opposite side) during unilateral (right-side) loadings reached almost 2500 εμ in the two-MDI models and were also higher in the splinted than in the unsplinted model, probably due to the rigid bar transferring more loads to peri-implant tissue than the attachments with the yellow PEEK inserts in the metal housings in the single-unit two-MDI model, whose small resiliency probably allowed more load to be transferred to a denture bearing area. At the same time, bilateral loadings with the same force (300 N) elicited significantly lower peri-implant strains, not exceeding 2000 εμ, highlighting the importance of bilateral chewing with implant-supported overdentures. Splinting status in combination with the extent of loading force did not have a significant effect on the peri-implant strains (*p* > 0.05). The findings of this study clearly point out the importance of bilateral chewing in subjects wearing overdentures supported by the two Ti-Zr mini-implants. Under the bilateral chewing scenario, only two Ti-Zr MDIs could successfully support the mandibular overdenture in the real patient situation, no matter the splinting status; however, unilateral chewing could lead to high peri-implant strains and consequent unwanted peri-implant bone loss. Subjects having complete dentures as antagonists have lower chewing forces and therefore may benefit from only two Ti-Zr MDI-supported overdentures for a longer period in a clinical situation, especially when chewing bilaterally. However, in subjects with unilateral chewing habits and in those having high chewing forces (natural teeth or fixed partial denture in the maxilla), the two Ti-Zr MDI-supported overdentures should not be recommended, as bone overloads can lead to peri-implant bone loss due to too high peri-implant strains. Also, the two-MDI overdenture should not be recommended in subjects with clenching or bruxing habits, as they cannot control the extent of the applied force during sleeping [54]. Under the 150 N force and unilateral loadings, higher values in the splinted two-MDI model registered on the left side can be again attributed to more force transferred by the rigid bar to the left side during the right-side loadings than by the PEEK attachment in the titanium housings in the single-unit model. During anterior loadings, a bit higher peri-implant values registered in the single-unit two-MDI model on the left side may be due to small variations in the inclinations of incisal edges of the anterior denture teeth eliciting small variations in load transfers. Other factors should also be accounted for in predicting peri-implant stress and strains, such as a removable denture stability, thickness of oral mucosa under the denture, quality of a denture, artificial teeth cusp inclination, occlusal scheme, etc., which were not analyzed in the present “in vitro” study and need further research.

Jofree et al. [55] did not find an association between peri-implant marginal bone loss and the extent of bite forces during the 15-month observation period when two Ti90Al6V4 MDIs were splinted with a bar supporting the mandibular OD. They found significantly less peri-implant bone loss in the two-MDI bar-supported overdentures and addressed it for a better stabilization of the overdentures and less denture movements than in overdentures when two single-unit MDIs retained the overdentures. The “o” rings in metal housings allow more denture movements in the single-unit MDIs than when they are supported with a bar. However, new Straumann^®^ Optiloc Retentive System^®^ with a high-performance PEEK (Polyether ether ketone) matrix placed in the titanium housing offers better retention and stability to an overdenture, and less movements compared with the overdentures retained by “o” rings, thus the splinting effect with a bar was not so evident in this in vitro study. Also, in line with the results of this study are the results of some clinical observations when only a small amount of peri-implant bone loss was recorded over three or five years when only two mini-implants supported removable partial dentures together with the remaining patients’ teeth [56,57,58], which helped to additionally stabilize the partial removable dentures.

In the three-MDI models, peri-implant strains also increased with higher forces and unilateral loadings, showing similar values in the single-unit and splinted models. However, the highest microstrains during unilateral loadings of about 2000 εμ indicate a possibility of the clinical utilization of only three Ti-Zr mini-implants for the mandibular overdenture support with less precautions to be accounted for than when only two MDIs are used. This is in line with a recent clinical observational study when three or four Ti90Al6V4 mini-implants supported the mandibular overdenture and no significant differences were found in peri-implant marginal bone loss over five years between the three and four mini-implants [9]. Peri-implant strains closer to the applied force were higher; therefore, peri-implant strains were higher during anterior loadings (although less than 2000 εμ) in the midline MDI. High peri-implant strains were also recorded in the midline implant during unilateral loadings (although less than 2000 εμ). Although splinting showed significant effects, the differences between splinted and single-unit peri-implant strains were small and probably without any clinical importance in a real patient situation, as even maximum recorded strains did not surpass 2000 εμ.

In the four-MDI mandibular models, strains behaved in a similar way as in the three-MDI models, although with even lower peri-implant strains recorded than in the three-MDI model. As in the three-MDI situations, the four MDIs can be safely used in clinical practice. The peri-implant strains had the highest values during unilateral (right-side) loadings with 300 N forces, although not exceeding 2000 εμ. Maybe precaution should be taken only for unilateral loadings with the highest forces during unilateral chewing in the three- and four-MDI models. All other strains did not exceed 1500 εμ, and cannot interfere with a possibility of bone reparation. Although the MANOVA analysis with the peri-implant microstrains as dependent variables and loading position, loading force, and splinting status as factors (the three-factor MANOVA) showed that the model was significant (*p* < 0.001), it has no clinical significance in a real patient situation. Higher loading forces increased microstrain values (*p* < 0.001). The effect of the loading position was also significant (*p* < 0.01) since unilateral loadings elicited high microstrains, followed by bilateral and frontal loadings. As all microstrains were less than 2000 εμ, it is not likely that denture loadings will interfere with bone reparation, and it is not likely that marginal bone loss would happen due to overloading and peri-implant stress.

In brief, peri-implant microstrains in the three- and four-MDI models had lower values than in the two-MDI model (both in splinted and unsplinted models). Maximum peri-implant strain, in the single-unit three-MDI model, from the right posterior MDI peri-implant area loaded unilaterally with the 300 N was about 2000 εμ. The peri-implant strain values were lower in the midline MDI. The registered strains in the four MDIs were always lower than 2000 εμ, even under the highest forces and unilateral loading position. It is not likely that peri-implant bone resorption would occur in a real patient situation in the three- and four-MDI models, no matter of the splinting status. This is also in line with a recent longitudinal clinical study with mini-implants made of Ti90Al6V4 grade 5 alloy [9].

One study proved that local conditions are more important for promoting bone atrophy under the denture saddles than systemic factors [59]; therefore, we measured not only peri-implant strains, but also strains in the edentulous area. Microstrains registered from the edentulous areas (descriptive statistics) under the OD saddles in the two-, three-, and four-MDI models (unsplinted and splinted), presented in Figure 6 and in Appendix A (descriptive statistics), showed values below 2000 εμ. The highest value was obtained in the two-MDI model compared with the three- and four-MDI models. The four-factor MANOVA model (Appendix A) (dependent variables: posterior right and left edentulous area) revealed significant effects (*p* < 0.001). Higher loading forces elicited higher strains under saddles (*p* < 0.01) with the highest value during unilateral OD loadings in the two-MDI non-splinted model from the right-side edentulous area (*p* < 0.01); however, strains were below 2000 εμ. Splinting status elicited significant effects (*p* < 0.001; i.e., higher microstrains in the single-unit models). Microstrains from the posterior edentulous area in the three-MDI mandibular models, presented in Figure 6 and Appendix A, revealed that the registered values were below 1800 εμ even under the highest applied forces in the unsplinted MDI models. The same was with the four-MDI models, when the highest recorded microstrains in the unsplinted model were below 1600 εμ.

Generally, splinting of Ti-Zr mini-implants decreased strains in the posterior edentulous area under denture saddles. By increasing the number of implants, strains were also significantly reduced in the posterior edentulous areas under denture saddles. No significant differences found on the left side between the three- and the four-MDI models was attributed to higher microstrains on the right side during unilateral loadings (only right-side anterior loading was performed). However, splinting and increasing numbers of mini-implants would be of little clinical importance, as all the recorded strains, even in the two-MDI unsplinted model where the highest microstrains were recorded, were below the threshold when the resorptive changes in bone tissue can begin. Therefore, it is not likely that denture settling would promote bone atrophy, even in the unsplinted two-MDI model, as microstrains registered from the posterior edentulous areas were below 2000 εμ. The results concerning edentulous area strains recorded in this study are in line with the reduced amount of residual ridge atrophy reported in clinical follow-up studies when implant-supported overdentures were observed and compared to conventional complete dentures [60,61,62,63].

The strength of this study is that the experiments covered the typical “real” mouth situation, including those with high chewing forces. However, limitations also exist, as this study covers only one of the possible situations in a real patient situation. The strain values may differ depending on the shape of each individual mandible. So, further research must be addressed in testing stress and strains in different mandibles with reduced bucco-lingual widths, but of different shapes. In clinical practice, different anatomies of the residual ridges and different bone densities and bone architectures can be found, as well as different thicknesses and consistencies of the attached mucosa of a denture bearing area. Moreover, small inclinations between the inserted mini-implants may exist in some patients due to different topography of residual ridges preventing absolute parallelism during their insertion, while in this study, Ti-Zr mini-implants were placed parallel to each other. All listed factors need further investigation.

## 5. Conclusions

Within the limitations of this study, the conclusion is that increasing numbers of mini-implants reduce peri-implant microstrains and those in the posterior edentulous area. By increasing the extent of loading forces, peri-implant microstrains increase almost linearly. Unilateral loading elicits the highest peri-implant microstrains on the loaded side irrespective of splinting. Unilateral loadings with the highest forces (250 and 300 N) induced almost 3000 εμ peri-implant microstrains (on the loaded side) and 2500 εμ on the opposite side in the two-MDI models, which can interfere with peri-implant bone reparation. Therefore, clinically, two mini-implants for the overdenture support can be used only in subjects with lower chewing forces and in those chewing bilaterally. Splinting only reduced strains in the posterior edentulous areas under denture saddles, but it is not of clinical importance, as the highest microstrain value in the posterior edentulous area did not exceed 2000 εμ, while the majority of values were beyond 1500 εμ, thus confirming that implant-supported overdentures minimize edentulous ridge atrophy. Further research must address models with different residual ridge shapes, bone densities, and thicknesses and consistencies of the keratinized mucosa.

## Figures and Tables

**Figure 1 jfb-15-00260-f001:**
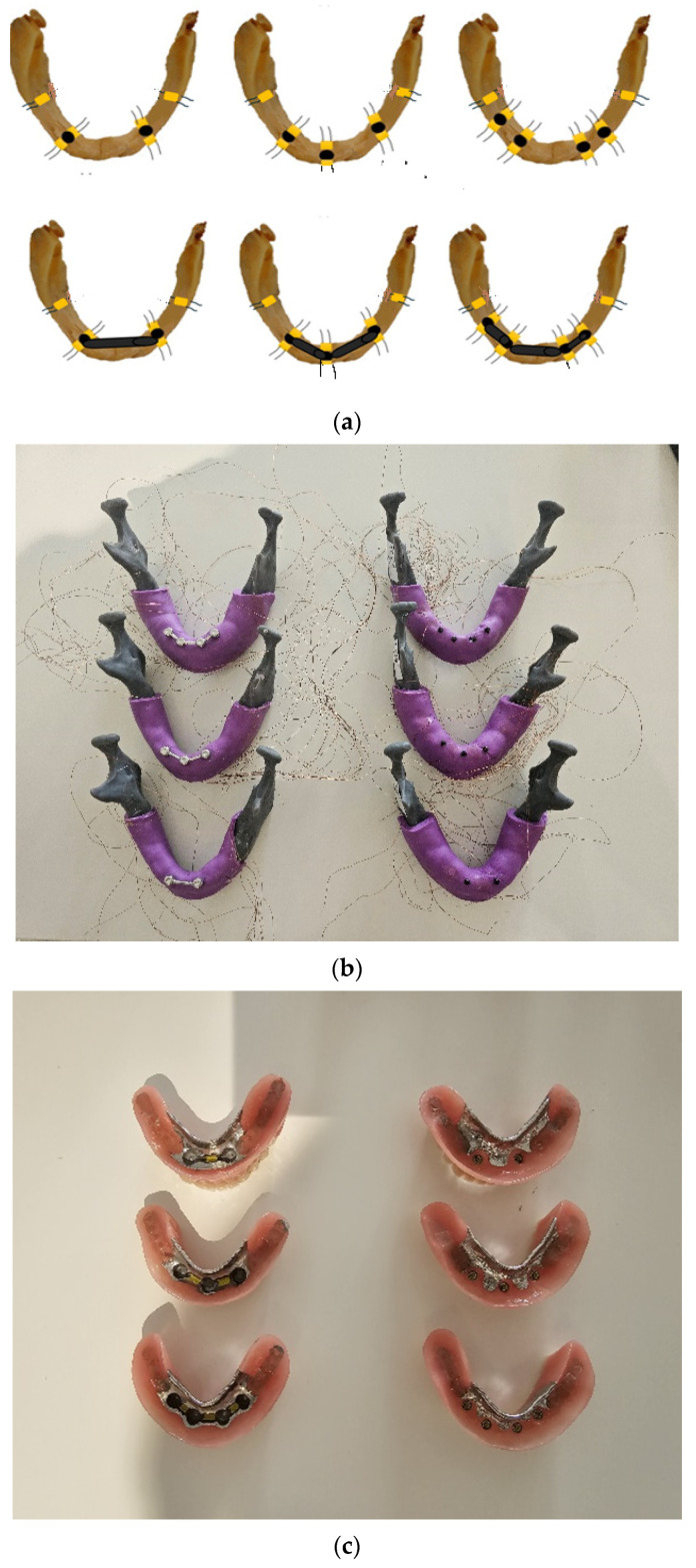
(**a**) Schematic drawings of the strain gauge positions in single-unit and splinted mini-implants; (**b**) mandibular models [splinted (**left** side) and single-unit MDIs (**right** side)] with wires from strain gauges (**b**) and the respective overdentures (**c**).

**Figure 2 jfb-15-00260-f002:**
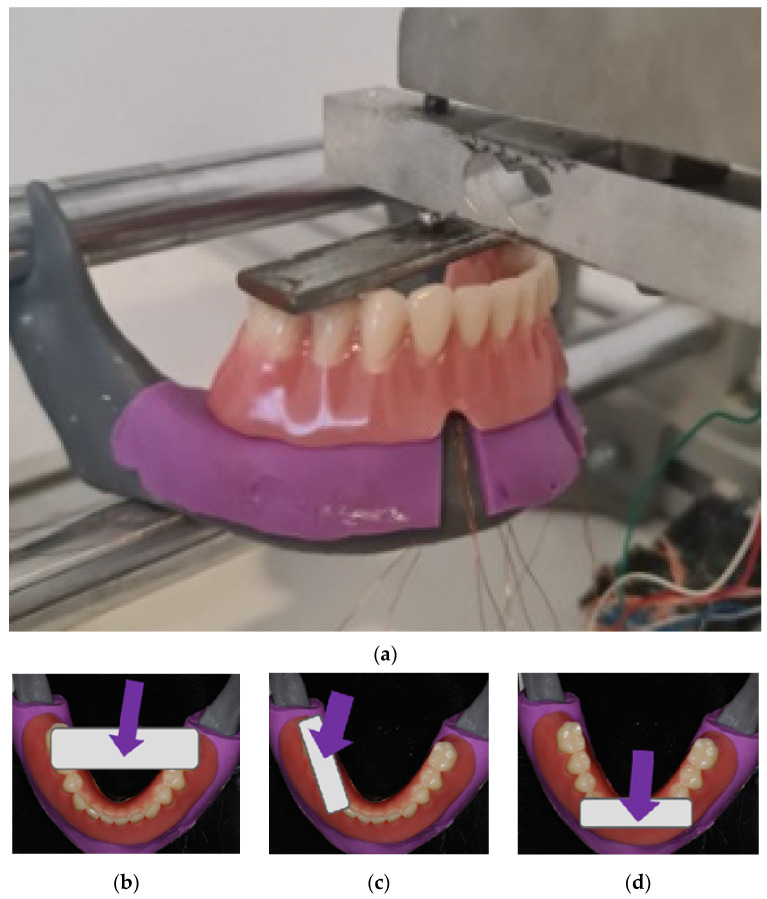
(**a**) Denture loading: the metal screw twisted to apply a pressure through the metal plate on the denture; (**b**) bilateral loading; (**c**) unilateral (**right** side) loading; (**d**) anterior loading.

**Figure 3 jfb-15-00260-f003:**
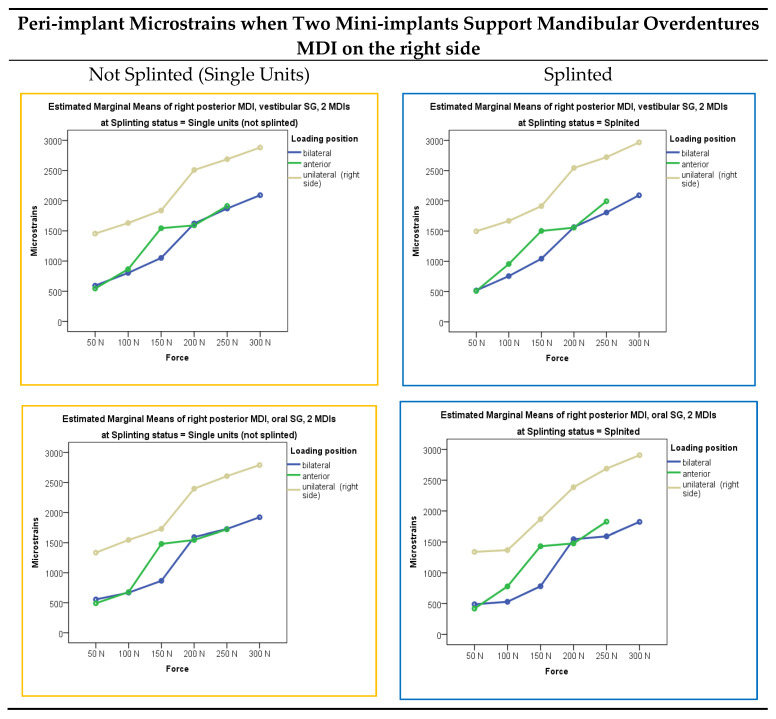
Peri-implant microstrains in the two-MDI models dependent on the loading force, loading position, and splinting status.

**Figure 4 jfb-15-00260-f004:**
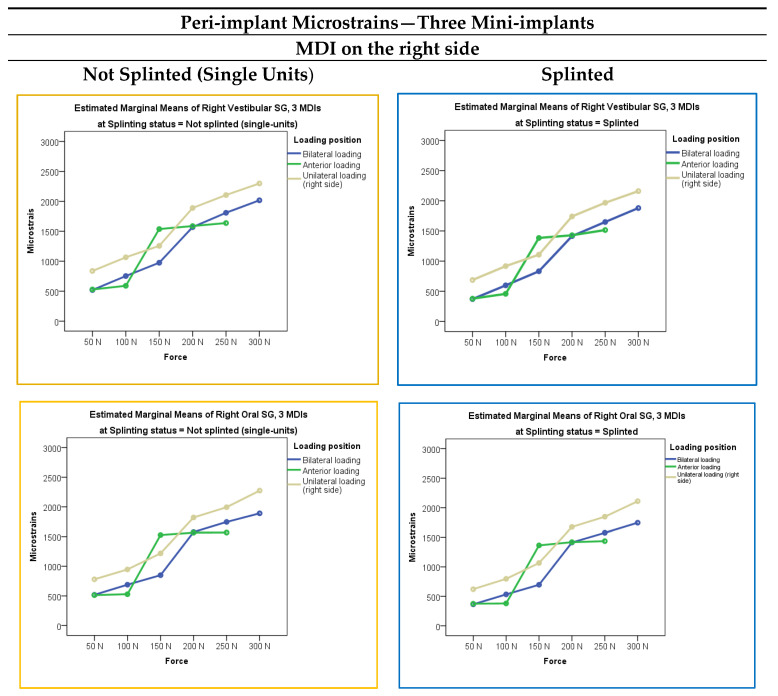
Peri-implant microstrains in the three-MDI models dependent on the loading force, loading position, and splinting status.

**Figure 5 jfb-15-00260-f005:**
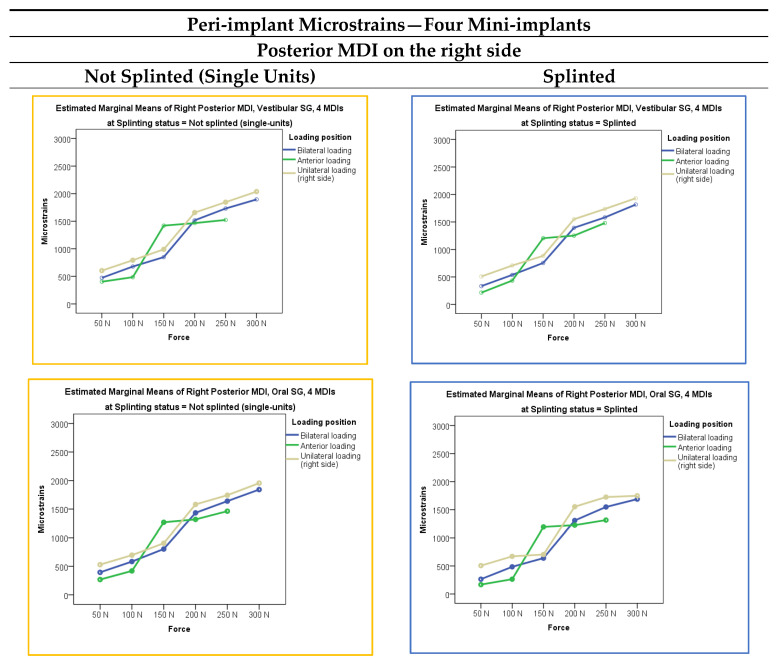
Peri-implant microstrains in the four-MDI models dependent on the loading force, loading position, and splinting status.

**Figure 6 jfb-15-00260-f006:**
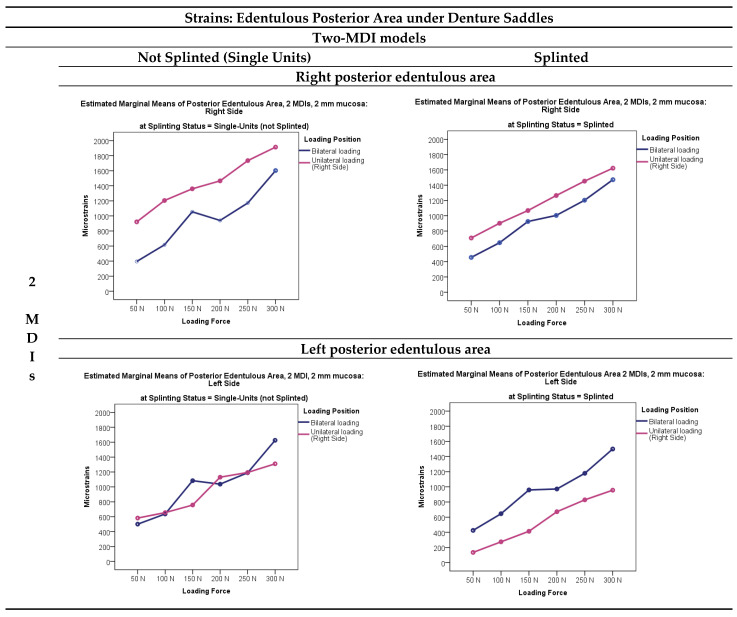
Microstrains registered from the right and left edentulous areas under the mandibular overdentures supported by single-unit or splinted MDIs supporting the overdenture, which was loaded in different loading positions (unilaterally–right side, bilaterally, and anteriorly) with different loading forces (50, 100, 150, 200, 250, and 300 N, respectively).

## Data Availability

The original contributions presented in the study are included in the article/Appendix A, further inquiries can be directed to the corresponding authors.

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
