# Peer review of "Effects of Loading Forces, Loading Positions, and Splinting of Two, Three, or Four Ti-Zr (Roxolid®) Mini-Implants Supporting the Mandibular Overdentures on Peri-Implant and Posterior Edentulous Area Strains"

_jfb, 2024, doi:10.3390/jfb15090260_

Round 1

Reviewer 1 Report

Comments and Suggestions for Authors

This manuscript entitled “Effects of Loading forces, Loading positions and Splinting of Two, Three or Four Ti-Zr (Roxolid®) Mini Implants Supporting the Mandibular Overdentures on Periimplant and Edentulous Area Strains” reports the effect of the number of mini-implants and the presence or absence of splints on bite forces is evaluated on special models. This study contains useful information for the use of mini-implants in edentulous patients. The content is interesting, but the figures are poorly organized and difficult to understand.

1.     In materials and methods, the hole width is mentioned as one mm narrower than the implant width. Isn’t it 0.1 mm?

2.     Please correct the font size on line 163, and 190.

3.     Not all figures are adjusted, and it is difficult to understand.

4.     The number of samples for each group is 2. Isn't the number of samples statistically insufficient?

Author Response

Reviewe1.    1. In materials and methods, the hole width is mentioned as one mm narrower than the implant width. Isn’t it 0.1 mm? Answer: yes, it is 0.1 mm, now we hope better explained in the text, in yellow color

2.     Please correct the font size on line 163, and 190. Answer: done (in yellow color in the text)

3.     Not all figures are adjusted, and it is difficult to understand. Answer: We have explained all what we have done in more details in already published manuscripts, and we provided references. We explained how we made the experiments, we added some more explanations now.  We also adjusted the photos in the text. The fact that the photos were not adjusted maybe happened because we sent all the photographs to the journal separately from the text of the manuscript, and probably the technical editor made such formatting.

4.     The number of samples for each group is 2.  Isn't the number of samples statistically insufficient?  

Answer: The number of samples was 15, now it is better explained, lines 166-169. There were 2 groups (splinted or not-splinted)  of 3 models  (with 2, 3 or four mini implants inserted in each model). The overdenture (OD) was made for each model and the OD was loaded I 3 positions with 6 different forces. During the OD loading strains were transferred to the periimplant and edentulous area, which were recorded by strain gauges. Each loading was repeated 15 times for 2 seconds for each loading position and force in each model and these values were taken for statistical analysis.

We made a total of (2x3=6 models) x 3 loading positions (18) x 6 different loading forces for bilateral and unilateral loadings (18x12=216) + 5 loading forces for anterior loadings (18x5=90)- 90+216= 306;

Finally: Each loading was repeated 15 times = 306 x 15 = 4590 results for each straingauge.

REVIEWER 2

1. In each hole (2.3 mm wide), a mini implant (2.4 mm wide and 10 mm long; Straumann® Mini Implant, Institute Straumann AG, Basel, Switzerland) with a neck height of 2.8 mm was inserted." Does the insertion of a 2.4 mm diameter implant into a 2.3 mm hole inherently cause stress in the bone surrounding the implant, and was this stress measured?  

  • Answer: Mini implants are compressive implants and the bur (drill) diameter for bone preparation in the original set is 0.1 mm narrower than the implant diameter to achieve a good primary stability in a real bone. It gently compresses the bone. The mini implants in this study were screwed into the 0.1 narrower holes in the models which mimicked the D2 density bone (not into the real bone). A 0.1 mm narrower hole is proposed by the manufacturer, as the set for insertion into the real bone has 0.1 mm narrower bur. It compresses the bone a little bit to achieve the optimal primary stability of the implant. This compression was not measured; we measured the periimplant strains caused by the overdenture (OD) loadings. The OD was retained by Ti-Zr mini implants and was loaded by different loading forces at different positions. The loads were transferred to the implants and periimplant bone tissue. Periimplant strains were measured, as well as strains from the posterior edentulous areas under the denture saddles. The minimum of 1 mm (or more) bone surrounded each implant, and we measured periimplant strains caused by load transfer to the periimplant “bone”, as strain gauges were glued as close as possible to the implant at the implant-model interface. The periimplant strains which were elicited by load transfer to the periimplant bone may be responsible for possible resorptive changes around implants (bone loss around implants). If it surpasses 3000 microstrains, or if strains repeat frequently, then > 2000 microstrains may interfere with bone reparatory mechanisms. Therefore we measured periimplant strains using strain gauges glued on the models as close as possible to the implant-“bone” interface. We also measured strains from the posterior edentulous areas under the denture saddles.

2. The borders of the images in the results section are inconsistent in size and need to be adjusted 

Answer: we made adjustments, but it was not our fault, as we had to upload photos separately and probably the technical editor formatted the photographs in the text. We adjusted the photos now.

3.

3、In Table 1, under "Two Mini-implants Supporting Mandibular Overdenture," for "MDI on the left side, anteriorly, and unilaterally (right side) with loading forces 150 N," there is a significant difference between the "Not Splinted (Single Units)" and "Splinted" results. However, this difference was not explained in the results section. Please explain the reason.

Answer: The supplementaryTable 1 shows only descriptive statistics of periimplant strains!????? Which Table did you mean? By mistake the 150 N force was written in the discussion of this manuscript, instead of the highest force, which was 300 N, when reporting microstrains from the edentulous area. We corrected that and we explained that although the difference was significant, it is probably of no clinical relevance. The highest strains from the edentulous area recorded in the 2-MDI unsplinted model under 300 N unilateral force on the right side were about 2000 microstrains; only higher strains than 3000 may interfere with the bone reparatory mechanisms, or repetitive strains higher than 2000. As all other strains were lower, this confirmed the fact that implant insertion reduces bone atrophy in the edentulous area under saddles compared to complete dentures. Splinting induced lower forces in the edentulous area. Average forces in implant retained overdentures are about 150 N. The 300 N high forces are rare and may happen only occasionally, only in bruxers (who are recommended to take out the denture during night) or in those with natural teeth or a fixed partial denture in the maxilla. The OD wearers are recommended to chew bilaterally.

To be more clearer, in the discussion, when strains were mentioned, we added periimplant strains when it was referred to them, and posterior edentulous area strains when we were discussing that matter.

4、Some references in the references section have incorrect formatting  - Answer corrected 

Reviewer 2 Report

Comments and Suggestions for Authors
  • 1"In each hole (2.3 mm wide), a mini implant (2.4 mm wide and 10 mm long; Straumann® Mini Implant, Institute Straumann AG, Basel, Switzerland) with a neck height of 2.8 mm was inserted."Does the insertion of a 2.4 mm diameter implant into a 2.3 mm hole inherently cause stress in the bone surrounding the implant, and was this stress measured?

    2The borders of the images in the results section are inconsistent in size and need to be adjusted.

    3In Table 1, under "Two Mini-implants Supporting Mandibular Overdenture," for "MDI on the left side, anteriorly, and unilaterally (right side) with loading forces 150 N," there is a significant difference between the "Not Splinted (Single Units)" and "Splinted" results. However, this difference was not explained in the results section. Please explain the reason.

    4Some references in the references section have incorrect formatting    

Author Response

  • 1、"In each hole (2.3 mm wide), a mini implant (2.4 mm wide and 10 mm long; Straumann® Mini Implant, Institute Straumann AG, Basel, Switzerland) with a neck height of 2.8 mm was inserted." Does the insertion of a 2.4 mm diameter implant into a 2.3 mm hole inherently cause stress in the bone surrounding the implant, and was this stress measured? Answer: Mini implants are compressive implants and the bur (drill) diameter for bone preparation in the original set is 0.1 mm narrower than the implant diameter to achieve a good primary stability in a real bone. It gently compresses the bone. The mini implants in this study were screwed into the 0.1 narrower holes in the models which mimicked the D2 density bone (not into the real bone). A 0.1 mm narrower hole is proposed by the manufacturer, as the set for insertion into the real bone has 0.1 mm narrower bur. It compresses the bone a little bit to achieve the optimal primary stability of the implant. This compression was not measured; we measured the periimplant strains caused by the overdenture (OD) loadings. The OD was retained by Ti-Zr mini implants and was loaded by different loading forces at different positions. The loads were transferred to the implants and periimplant bone tissue. Periimplant strains were measured, as well as strains from the posterior edentulous areas under the denture saddles. The minimum of 1 mm (or more) bone surrounded each implant, and we measured periimplant strains caused by load transfer to the periimplant “bone”, as strain gauges were glued as close as possible to the implant at the implant-model interface. The periimplant strains which were elicited by load transfer to the periimplant bone may be responsible for possible resorptive changes around implants (bone loss around implants). If it surpasses 3000 microstrains, or if strains repeat frequently, then > 2000 microstrains may interfere with bone reparatory mechanisms. Therefore we measured periimplant strains using strain gauges glued on the models as close as possible to the implant-“bone” interface. We also measured strains from the posterior edentulous areas under the denture saddles.

2、The borders of the images in the results section are inconsistent in size and need to be adjusted –Answer: we made adjustments, but it was not our fault, as we had to upload photos separately and probably the technical editor formatted the photographs in the text. We adjusted the photos now.

3、In Table 1, under "Two Mini-implants Supporting Mandibular Overdenture," for "MDI on the left side, anteriorly, and unilaterally (right side) with loading forces 150 N," there is a significant difference between the "Not Splinted (Single Units)" and "Splinted" results. However, this difference was not explained in the results section. Please explain the reason.

Answer: The Table 1 shows only descriptive statistics of periimplant strains!????? Which Table did you mean? By mistake the 150 N force was written in the discussion of this manuscript, instead of the highest force, which was 300 N, when reporting microstrains from the edentulous area. We corrected that and we explained that although the difference was significant, it is probably of no clinical relevance. The highest strains from the edentulous area recorded in the 2-MDI unsplinted model under 300 N unilateral force on the right side were about 2000 microstrains; only higher strains than 3000 may interfere with the bone reparatory mechanisms, or repetitive strains higher than 2000. As all other strains were lower, this confirmed the fact that implant insertion reduces bone atrophy in the edentulous area under saddles compared to complete dentures. Splinting induced lower forces in the edentulous area. Average forces in implant retained Overdentures are about 150 N. The 300 N high forces are rare and may happen only occasionally, only in bruxers (who are recommended to take out the denture during night) or in those with natural teeth or a fixed partial denture in the maxilla. The OD wearers are recommended to chew bilaterally.

To be more clearer, in the discussion, when strains were mentioned, we added periimplant strains when it was referred to them, and posterior edentulous area strains when we were discussing that matter.

4、Some references in the references section have incorrect formatting  - corrected 

Round 2

Reviewer 1 Report

Comments and Suggestions for Authors

I have understood your explanation. However, you mentioned the mandible model was obtained from one patient, so the sample number was one in this study. That one sample was tested 15 times. It is expected that the measured values ​​may differ depending on the shape of each individual mandible. I understand that one sample was tested 15 times in this study, but I think it would have been better to increase the number of samples.  Please add a few comments about it in the discussion.

Author Response

REVIEWER 1. I have understood your explanation. However, you mentioned the mandible
model was obtained from one patient, so the sample number was one in
this study. That one sample was tested 15 times. It is expected that the
measured values ​​may differ depending on the shape of each individual
mandible. I understand that one sample was tested 15 times in this
study, but I think it would have been better to increase the number of
samples.  Please add a few comments about it in the discussion.

Answer:

Added, in the end of Discussion, as well as in Conclusions in yellow colour

The strength of this study is that the experiments covered the typical "real" mouth situation, including those with high chewing forces. However, the limitations also exist, as the study covers only one of the possible situations in the real patient situation. The strain values ​​may differ depending on the shape of each individual mandible. So further research must be addressed in testing several different mandibles having reduced bicco-lingual width, but of different shapes. In clinical practice different anatomies of the residual ridges, different bone densities and bone architectures can be found, as well as different thickness and consistencies of the attached mucosa of a denture bearing area. Moreover, small inclinations between the inserted mini-implants may exist in some patients due to different topography of residual ridges preventing absolute parallelism during their insertion, while in this study Ti-Zr mini-implants were placed parallel to each other. All listed factors need further investigation.

  1. Conclusions

Within the limitations of this study the conclusion is that increasing numbers of mini implants reduces periimplant microstrains and those in the posterior edentulous area. By increasing the extent of loading forces periimplant microstrains increase almost linearly. Unilateral loading elicits the highest periimplant microstrains on the loaded side irrespective of splinting. Unilateral loadings with the highest forces (250 and 300 N) induced almost 3000 εμ periimplant microstrains (on the loaded side) and 2500 εμ on the opposite side in the two-MDI models, which can interfere with periimplant bone reparation. Therefore, clinically two mini implants for the overdenture support can be used only in subjects with lower chewing forces and in those chewing bilaterally. Splinting reduced only strains in the posterior edentulous areas under denture saddles, but it is not of clinical importance, as the highest microstrain value in the posterior edentulous area did not exceed 2000 εμ, while majority values were beyond 1500 εμ, thus confirming that implant supported overdentures minimize edentulous ridge atrophy. Further research must be addressed to models with different residual ridge shapes, bone densities and thickness and consistencies of the keratinized mucosa.

Reviewer 2 Report

Comments and Suggestions for Authors

The author has made substantial revisions to the article. Regarding the issues raised in the first round of review, there was an error in the description of the third issue. It is not Table 1, but rather Figure 3 under "Two Mini-implants Supporting Mandibular Overdenture," where for "MDI on the left side," with loading forces (150 N) on the anteriorly, and unilaterally (right side), there is a significant difference between the "Not Splinted (Single Units)" and "Splinted" results. However, this difference was not explained in the results section. Please provide an explanation for this phenomenon.

Author Response

The author has made substantial revisions to the article. Regarding the
issues raised in the first round of review, there was an error in the
description of the third issue. It is not Table 1, but rather Figure 3
under "Two Mini-implants Supporting Mandibular Overdenture," where for
"MDI on the left side," with loading forces (150 N) on the anteriorly,
and unilaterally (right side), there is a significant difference between
the "Not Splinted (Single Units)" and "Splinted" results. However, this
difference was not explained in the results section. Please provide an
explanation for this phenomenon.

  • This is mentioned in the results section,
  • ………..variable Loading Position in the 2-MDI models are presented in Supplementary Table 4. Periimplant microstrains during loading of the mandibular OD in the three loading positions (bilaterally, anteriorly and unilaterally on the right-side) were significantly different from each other, except for the periimplant microstrains from the left oral SG when strains during bilateral loadings did not differ significantly from anterior loadings. On the left side, under the 150 N force, there was a significant difference between the "Not Splinted (Single-Unit)" and "Splinted" models considering the registered microstrain values during anterior and unilateral loadings.

  • And explained in the discussion: Lines 412-419 in yellow colour:

Under the 150 N force and unilateral loadings higher values in the splinted two-MDI model registered on the left side can be again attributed to more force transferred by the rigid bar to the left side during the right-side loadings than by the PEEK attachment in the titanium housings in the single-unit model. During anterior loadings a bit higher periimplant values registered in the single-unit two-MDI model on the left side may be due to small variations in the inclinations of incisal edges of the anterior denture teeth eliciting small variations in load transfers.
